# Towards inclusive authorship: Analyzing author representation in *PLOS Global Public Health* front matter content

**Esme Supriya Gupta Longley**, **Shashika Bandara***

Department of Global and Public Health, School of Population and Global Health, McGill University, Montreal, Québec, Canada

* shashika.bandara@mail.mcgill.ca

## Abstract

Underrepresentation and lack of inclusion of Global South researchers have been key shortcomings in global health publications. This has contributed to epistemic injustice in global health and impacted evidence informed policymaking. PLOS Global Public Health (GPH)was launched in 2021 with the goal of charting a new path towards equity, diversity and inclusion in global health publications. The journal also invited independent assessments of its progress. This study analyses commissioned 136 front matter content (opinions, reviews, and essays) and a total of 878 authors published in PLOS GPH between October 2021 and December 2024. Using publicly available data from the journal website and online profiles, we examined authorship representation based on World Bank country income classification, gender, and Indigeneity. Additionally, we examined article content in terms of country focus and topics covered. We inferred gender by reviewing public profiles for gendered prefixes and pronouns and when unavailable by using genderize.io. We analyzed for Indigeneity by reviewing authors' public profiles. Our results indicate that 609 of 878 (69%) of authors for the commissioned content were affiliated with high income countries. Under gender representation, 403 of 878 (46%) authors identified as women compared to 471 of 878 (54%) as men. Only 7 of 135 (5%) first authors and 6 of 117 (5%) senior authors publicly identified as Indigenous. While most articles had a global focus (78 of 136, or 57%), 46 of 136 (34%) focused on the Global South, and 12 of 136 (8%) on the Global North. Global South affiliated authors were better represented in articles pertaining to the Global South, comprising on average 43% of authorship compared to an overall average of 30%. To advance equity, journals should commission more content from Global South authors and actively invite contributions from Indigenous and gender-diverse authors on topics relevant to their communities.

**Data availability statement:** We have made the authorship data analysis available at https://osf.io/3fmun/files/osfstorage/682e778654e4a48751f07e21.

**Funding:** The authors received no specific funding for this work.

**Competing interests:** I have read the journal's policy, and the authors of this manuscript have the following competing interests: Shashika Bandara is an academic editor at the PLOS Global Public Health Journal. Esme Supriya Gupta Longley has no competing interests to declare.

## Introduction

Underrepresentation and lack of inclusion of Global South researchers have been long-standing challenges in global health publications contributing epistemic injustice and sustaining neo-colonial models of knowledge production [1–6]. Global health journals have also reckoned with the need for better inclusion policies in terms of gender, race, disability, and other non-dominant identities [1,2,4,7–9]. Parachute or helicopter research, high article processing charges (APCs), Global North centric funding structures, lack of journal policies on inclusive authorship, lack of support to overcome language barriers are some of the key barriers which have contributed to inequitable authorship practices in global health [4,5,8,10,11]. These systemic barriers have contributed to researchers from High-Income Countries (HICs) or dominant social groups often being designated as lead authors and in turn being recognized as experts in countries or settings that they do not have lived experience in [1,4,11]. Adithi Iyer, examining authorship in *Lancet Global Health*, captured this inequity, writing "it appears that 'disadvantaged populations' continue to suffer not only from under-representation, but also a plethora of missed opportunities to contribute to global health research," [8]. Epistemic injustice in global health knowledge production, in turn influence global and national policies, affecting the health and wellbeing of communities [1,4,11].

In the last decade, recognizing increasing calls to action, academic journals have taken measures to strengthen inclusive authorship policies [1,7,12,13]. Examples of journal efforts include the PLOS policy on inclusion in global research [14], a growing call for author reflexivity statements [15–17], *Lancet Global Health* detailing its planned efforts to "making global health research, publishing, and practice a more equitable and effective space," [18] *Springer Nature* family journals improved its open access policy to waive APC for over 70 low income and lower-middle income countries [19], and *Health Policy and Planning* committing to at least 50% of articles including a first or senior author from a low and middle income country [20]. However, stronger, intentional and continuous efforts are needed to progressively strengthen epistemic justice in global health. Seye Abimbola, based on his research on epistemic justice, described the ideal version of knowledge production being "local people writing about local issues for local audiences," with centering of local experts as an essential component [3].

Recognizing the existing inequities in knowledge production and the need to intentionally address these challenges, *PLOS Global Public Health* (PLOS GPH) was launched in 2021 with an aim to chart a new path towards equity, diversity and inclusion in global health [21]. According to the journal's two year analysis PLOS GPH has published articles from 85 corresponding authors with almost half coming from the Global South [22]. In its two-year update, PLOS GPH's editorial board committed further to "support authorship models that are more inclusive, equitable, self-reflexive," [22]. The editorial board further committed to avoid platforming "the same Global North voices" and intentionally seeking out Global South, Indigenous scholars and practitioners/activists with lived experiences, as a means to intentionally shift the

inequitable status quo in global health [22]. The editorial board also requested to support their efforts towards inclusion via independent assessments of its progress [21].

Therefore, in this study we examined the authorship representation among the commissioned or 'front matter' content (opinions, reviews and essays), of PLOS GPH. Among commissioned content of PLOS GPH, we examined country income category, gender, and Indigeneity representation in authorship. We also analyzed the subject matter focus of these articles. Our goal in conducting this study is to support the journal in its efforts to be accountable and build a more inclusive global authorship. The findings and discussion of this study are relevant to all global health journals in their efforts to improve inclusive and equitable publication practices.

## Methods

In this study, we conducted a) an authorship analysis and b) article content focus analysis of published opinion, review, and essay articles in PLOS GPH from its launch in October 2021 to December 31, 2024. We focused on commissioned or front matter content as these are the types of articles that the editorial board has most control over. We excluded all other article types from this analysis. We analyzed a total of 136 articles and 878 authors.

Under authorship representation analysis income category, gender, and Indigeneity of each author. While we did not contact authors for this information, we used publicly available data from PLOS GPH journal and publicly available affiliation data online when necessary. To identify the representative countries, we used the location of primary institutional affiliation as a proxy for represented country and region. In our analysis, we used World Bank's 2024 country income-based categories to group representative countries [23]. These categories are HIC, Upper-Middle-Income Country (UMIC), Lower-Middle-Income Country (LMIC), and Low-Income Country (LIC). For example, an author with a primary affiliation at McGill University would be considered as representing Canada and would be categorized under the HIC category.

For article content focus analysis, we examined the country or region the article content was focused on and the main subject matter focus (e.g., infectious diseases). We describe both data collection and analytical approach in detail below.

### Data collection and analysis

**Authorship representation information.** We collected the data for the location (country) of every author's primary institutional affiliation. Using this data, we first categorized authors based on countries represented (via institutional affiliation). Then using the World Bank country income classification, we first categorized all authors into two main categories: Global North (HIC), or Global South (including UMIC, LMIC and LIC). When multiple affiliations were listed for one author, we used the first affiliation listed as proxy and other affiliations were noted separately. There were no group or consortium authorship in the articles analyzed.

For first and senior authors, who often lead the writing and framing of commissioned opinion, review and essay articles we conducted an additional layer of analysis. In addition to primary institutional affiliations, for first and senior authors we also examined the secondary affiliations to determine additional countries represented. Using the collected authorship data we performed two types of calculations: a) the proportion of authors by country income category - for all authors included, then for first and senior authors b) the average proportion of authorship based on country income category for each type of front matter content (e.g., essay, opinion).

For gender analysis of authorship, we used a two-step approach. First, for all authors, we conducted a manual review of public information, such as gendered prefixes (e.g., Ms., Mr.) and pronouns from publicly available profiles. These online sources include biographies hosted on university or institute websites and unrestricted profiles in social media platforms, such as LinkedIn, Bluesky, and X. Second, for authors whose gender details could not be identified by the first step, which was 74 of 878 authors, or 8.4%, we used the genderize.io tool to infer each author's gender based on the first name [24]. This tool has been utilised in similar previous studies [25]. Inferences for gender based on the first name were only accepted if the probabilistic certainty score was ≥ 0.90. The genderize.io tool was only used as a final resort, since

non-binary identities are not recognised by the algorithm and non-Western names are inferred at a lower quality rating [26]. Gender was recorded as "unknown" if an inference was not possible. In the analysis, authors with a gender marked as "unknown" were excluded from the analysis due to their small sample size. For gender analysis we calculated the proportion of authorship based on available gender identities. We used these calculations to also summarise the number of articles based on the percentage of authors identifying as women.

To identify if any authors were Indigenous, we searched publicly available online profiles for terms indicating self-identified status as Indigenous (e.g., "Indigenous," "Aboriginal," "First Nations," "Inuit," "Native American"). Similar to gender analysis, these publicly available online sources include biographies hosted on university or institutional websites and unrestricted profiles on social media platforms, such as LinkedIn, Bluesky, and X. Given the limited data, we report the data we collected without additional calculations.

Table 1 summarizes the analytical approach and key categories related to authorship representation.

We only used publicly available data for our analysis and did not contact authors with questionnaires. Thus, based on established ethical guidelines, this study did not require approval from an ethical review board. Furthermore, due to none of authors of the analyzed articles publicly identifying as persons with disabilities, we did not conduct an analysis on the representation of persons with disabilities.

All the data used for this analysis and the key analyses are included uploaded as an Excel File to OSF. The data file can be accessed at https://osf.io/3fmun/files/osfstorage/682e778654e4a48751f07e21.

**Article content scope.** Using explicitly stated geographic focus of articles, we identified the primary geographic focus, be it a country (e.g., India) or a global region (e.g., African region). In articles that explicitly mentioned multiple geographic regions (e.g., South America, Africa), we included all the regions mentioned in the article, as its geographic focus areas. For articles that focused on globally relevant topics without explicitly noting a country or global region, we categorized those articles under the 'global' label. Using the identified countries of focus of the article, we also identified the income level classification of each country(ies) of focus. For this categorization, similar to the authorship information section, we used World Bank Income classification (i.e., HIC, UMIC, LMIC, LIC). Using this data, we calculated average proportion authorship affiliation by the article's country income category focus. Our goal was to analyze whether the authorship of the article was relevant to the countries they wrote about. For example, we wanted to examine if a higher proportion HIC authors were writing about UMIC, LMIC, or LICs and vice versa.

In terms of subject matter focus of the article, we assigned themes based on relevance of the subject matter. For the purposes of this research, considering the focus areas of all articles, we generated the following themes for content focus analysis: infectious diseases (e.g., HIV/AIDS), non-communicable diseases (NCDs) (e.g., diabetes), health infrastructure and delivery (e.g., health systems, healthcare delivery), and research partnerships (e.g., bilateral partnership strengthening). S1 Table includes a comprehensive list of the main focus of all articles analyzed listed under each of the four thematic categories noted above. In addition to descriptive data reporting, in our calculations, we examined the average

**Table 1. Key analytical approaches and resulting categories in authorship representation analysis.**

| Type of representation assessed | Data analysis approach | Key categories based on available public data |
| --- | --- | --- |
| *Country (Institutional affiliation based)* | Publicly available institutional affiliations of authors as proxy to identify countries and determine country income category | World Bank country income classification categories (HIC, UMIC, LMIC, LIC) |
| *Gender* | Gender inferred from the manual review of publicly available data and genderize.io algorithm (≥0.90 confidence threshold) | Man, Woman, Non-binary, Unknown |
| *Indigenous* | Public profile analysis of authors for self-identification of Indigenous status | Indigenous, non-Indigenous |

proportion of authorship based the country income category (e.g. HICs) for each of the subject matter themes (e.g., infectious disease). Table 2 summarizes the data analysis approaches and key categories for the article content scope analysis.

## Results

Overall, based on our descriptive analysis of articles, from the total of 136 articles, 39 were review articles, 93 were opinion articles, and 4 were essays. There were 878 total individual authors. There were total of 136 first authors and a total of 118 senior authors.

### Authorship representation

**Representation based on country income categories.** We considered all the authors (n = 878), across all article types (n = 136). In terms of country income categories based on author affiliation, 609 of total 878 authors (69%) were affiliated with HICs, while the remaining 269 of 878 (31%) were from UMICs, LMICs, or LICs. Furthermore, 56 of the 136 articles (41%) had over 90% of authors affiliated with HICs. In contrast, 24 of the 136 articles (18%) had over 50% of their authorship affiliated with UMICs, LMICs, or LICs.

First (n = 135) and senior (n = 117) author represented across all commissioned articles were also largely dominated by HIC institution affiliated authors. One of 136 first authors (0.7%), was classified as undetermined due to affiliation with a global institution without regional institutional affiliation. One senior author of the 118 (0.8%) had an undetermined categorization due to a global organizational affiliation. Therefore, these two authors were removed from the first and senior authorship country income category-based analyses.

Among first authors, 103 of 135 (76%) were affiliated with institutions in HICs, and 32 of 135 (24%) were affiliated with UMICs, LMICs, or LICs (Fig 1). Among non-HIC first authors the affiliation data is as follows: 11 of the total 135 first authors (8%) were affiliated with UMICs, 12 (8%) with LMICs, and nine (71%) with LICs. Among senior authors, 82 of 117 (70%) were affiliated with HICs and 35 of 117 (30%) were affiliated with UMICs, LMICs, or LICs (Fig 1). Affiliation data of non-HIC affiliated senior authors is as follows: 10 of the 117 total senior authors (9%) were affiliated with UMICs, 20 (17%) with LMICs, and five (4%) with LICs. Notably, LICs were particularly underrepresented among senior authors at 4% (5 out of 117). Additionally, 18 of the 136 total articles analyzed had only one listed author, who was counted as only the first author, thus was not included in the analysis for senior authors.

We also examined secondary institutional affiliations for first and senior authors. When secondary affiliations were considered, HIC representation saw marginal decrease. Among first authors, 97 of the 135 (72%) were affiliated exclusively with HICs, while 26 of 135 (19%) were affiliated exclusively with LICs, LMICs, and UMICs. A total of six of the 135 first authors (4%) listed dual affiliations with a primary HIC institution and a secondary LIC, LMIC, or UMIC institution, while another six of the 135 (4%) had primary LIC, LMIC, or UMIC affiliations with a secondary HIC institution (Fig 1).

**Table 2. Key analytical approaches and resulting categories in article content analysis.**

| Type of content focus assessed | Data analysis approach | Key categories |
|---|---|---|
| *The main subject matter focus* | Content analysis to determine primary subject matter in terms of health topics | Infectious Diseases, NCDs, Health Infrastructure and delivery, Research Partnerships |
| *The country or global region focus* | Content analysis to determine the country of interest and country income category | Global, HIC, LIC, LMIC, UMIC |

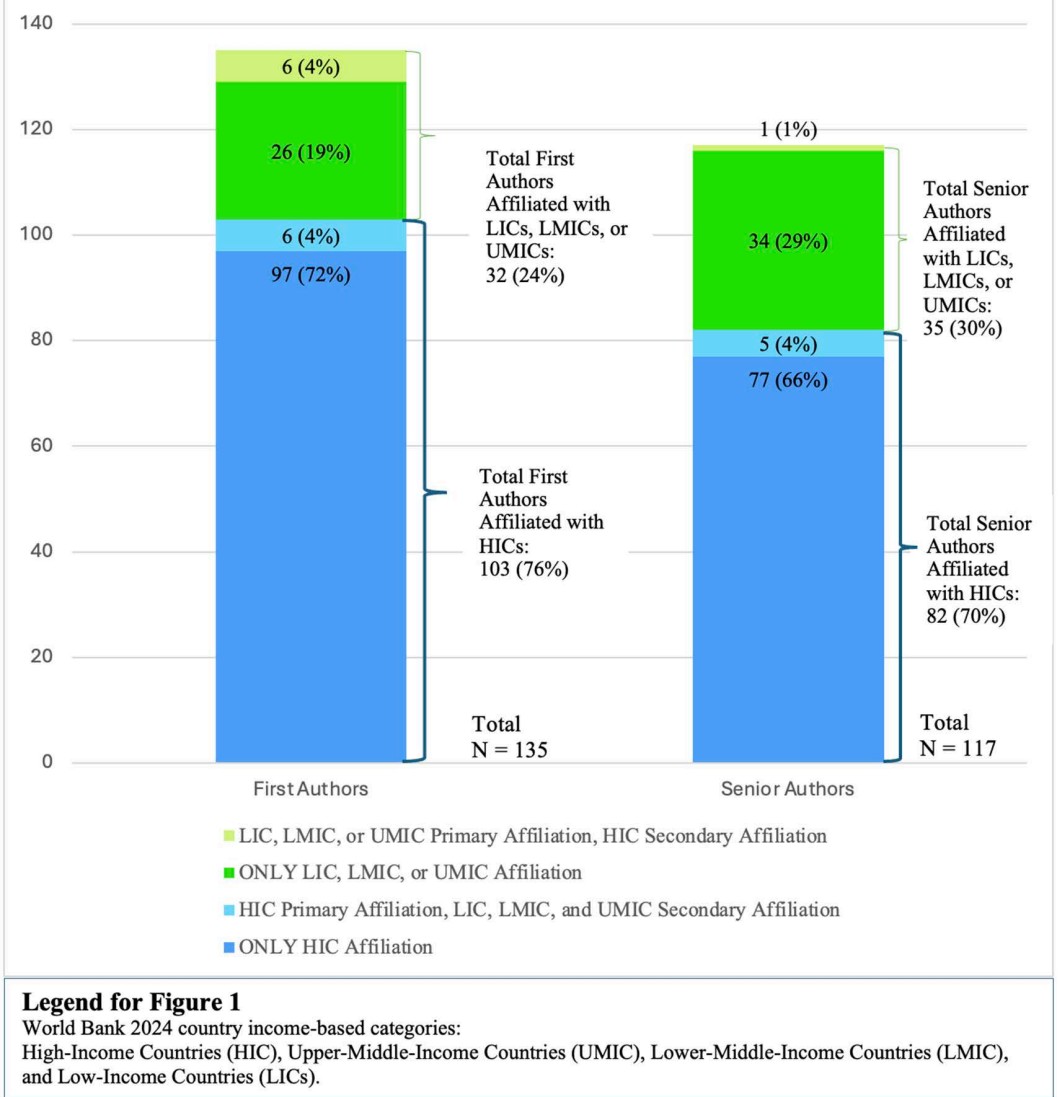

**Fig 1. First and senior author representation based on country income category using both primary and secondary affiliations.**

Similarly, considering secondary affiliations, among senior authors, 77 of the 117 (66%) had exclusive HIC affiliations, while 34 of 117 (29%) were exclusively affiliated with LIC, LMICs, or UMICs. Five of 117 (4%) had primary HIC affiliations with a secondary LIC, LMIC or UMIC affiliation, and one of 117 (0.9%) had a primary LIC, LMIC affiliation with a secondary HIC affiliation (Fig 1). We only conducted this analysis for first and senior authors, given their leading roles in writing and conceptualizing front matter content.

Considering article categories, essay articles saw the highest average representation of authors affiliated with LICs, LMICs, or UMICs (36%). Opinion articles also saw an above average representation (31%), but review type articles had below average representation from authors from LICs, LMICs, or UMICs (27%, compared to the 30% average). Table 3 summarizes average percentage of authorship based on country income category for each article type.

**Gender representation.** Using publicly available data (for 804 of 878 authors, or 91.6%) and genderize.io (for 74 of 878 authors, or 8.4%) we inferred gender identities across all 878 authors. Based on our analysis, 471 of the 878 (54%) identified as men, 403 of 878 (46%) identified as women, and three of 878 (0.003%) had an unknown gender. Among first

**Table 3. Article type by the average percentage of authors affiliated with LICs, LMICs, or UMICs.**

| Article Type | Average percentage of authors from LICs, LMICs, or UMICs |
|---|---|
| Essay | 36% |
| Opinion | 31% |
| Review | 27% |
| **Total** | **30%** |

authors, 56 of 135 (41%) identified as men and 80 of 135 (59%) as women. Among senior authors, 65 of 117 (56%) were men and 52 of 117 (44%) were women. Additionally, 17 of the 136 articles had a single listed author, who was counted as the first author and not as the senior author.

A total of 25 of 136 articles (18%) had 90–100% authors identifying as women. 54 of the 136 total articles (40%) had over 50% women authorship. Table 4 summarises the number of articles based on the percentage of authors identifying as women.

**Indigeneity.** Based on our analysis, seven of the 136 first authors (5%) publicly self-identified as Indigenous. Six of the 117 senior authors (5%) self-identified as being Indigenous.

## Article content focus and representation

**Article content focus - country income categories.** Based on the analysis, 78 of the total 136 articles (57%) were categorised as "global" as they addressed global topics (e.g., global health partnerships, WHO guidelines, knowledge systems) and did not explicitly focus on a specific global region or a country. Specifically, 21 of 39 review articles (54%), 54 out of 93 opinion articles (58%), and three out of four essay articles (75%) had a global focus. In contrast, 12 of the 136 articles (8%) specifically focused on HICs, while 46 of 136 (34%) focused on UMICs, LMICs, and LICs. Table 5 summarizes the focus of articles categorized by World Bank country income classification.

Articles whose content explicitly focused on LICs, LMICs, or UMICs saw the highest average representation of authors affiliated with LICs, LMICs, or UMICs (43%). On average, articles addressing 'global' topics saw a 25% representation from authors affiliated with institutions in LICs, LMICs, or UMICs. Articles focused on HICs, there was a substantially lower (11%) representation from authors affiliated with LICs, LMICs, or UMICs. See Table 6 for the summary of article focus based on country income category and authorship from LICs, LMICs, or UMICs.

**Article content focus—Subject matter.** Among the articles, the most topics were related to health infrastructure and delivery (59 of the 136 articles, or 43%). Research partnerships focus was second (46 of 136, or 34%), with infectious

**Table 4. Number of articles based on the percentage of authorship inferred to be women.**

| Percentage of women authors | Number of Articles (n total = 136) | Percentage of total articles |
|---|---|---|
| 0-10% Women authors | 20 | 15% |
| 10-20% Women authors | 6 | 4% |
| 20-30% Women authors | 12 | 9% |
| 30-40% Women authors | 15 | 11% |
| 40-50% Women authors | 29 | 21% |
| 50-60% Women authors | 8 | 6% |
| 60-70% Women authors | 10 | 7% |
| 70-80% Women authors | 9 | 7% |
| 80-90% Women authors | 2 | 1% |
| 90-100% Women authors | 25 | 18% |

**Table 5. Article content focus summary—Country income categories.**

| Type | Global | HICs | UMICs, LMICs, and LICs | Total |
|------|--------|------|------------------------|-------|
| *Review* | 21 | 5 | 13 | **39** |
| *Opinion* | 54 | 7 | 32 | **93** |
| *Essay* | 3 | 0 | 1 | **4** |
| **Total** | **78 (57%)** | **12 (8%)** | **46 (34%)** | **136** |

**Table 6. Article focus based on country income categories and authorship affiliation.**

| Article focus based on country income category | Average percentage of authors from LICs, LMICs, or UMICs |
|-----------------------------------------------|----------------------------------------------------------|
| LIC, LMIC, or UMIC | 43% |
| Global | 25% |
| HIC | 11% |
| **Total** | **30%** |

disease focus (23 of 136, or 17%) at third. NCDs were the least covered topic, with only eight of 136 (6%) focusing on this topic. Table 7 summarizes subject matter focus among articles analyzed.

In terms of subject matter focus, authors from HICs showed strong representation across all topics including NCDs, health infrastructure and delivery, research partnerships, and infectious diseases. Notably, articles focused on NCDs exhibited the highest proportion of HIC-affiliated authors (79%). Fig 2 provides an overview of authorship representation per different subject matter focus categories.

## Discussion

Lack of diversity in authorship in terms of Global South authors and authors representing other non-dominant groups such as Indigenous and gender diverse communities has been recognized [27–29]. This examination of front matter content of PLOS GPH highlights opportunities for PLOS GPH to strengthen their efforts in improving diverse representation in their authorship, to ensure equity and counter epistemic injustice in global health.

PLOS GPH's strong commitment to improving diversity, equity and inclusion can be observed from the inception of this journal. As of November 2023, PLOS GPH had 50–50 Global South – Global North representation in their editorial board including academic and section editors [22]. After two years of publishing (by November 2023), PLOS GPH has featured authors from 85 countries, across research articles, with almost half of them from the Global South [22].

In specifically examining authorship of commissioned content (opinion, review and essay articles), we assessed the progress made by PLOS GPH towards its own goal of committing to genuine diversity of authors and checking if the journal is platforming the same Global North voices. Our results indicate that, within commissioned content PLOS GPH needs to bring in more authors from the Global South. As of December 2024, 609 of 878 (69%) of all authors had primary affiliations in HICs. Additionally, 103 of 135 (76%) of first authors and 82 of 117 (70%) of senior authors were from HIC-affiliated

**Table 7. Article type categories based on subject matter focus.**

| Article Subject | Opinion | Review | Essay | Total |
|-----------------|---------|--------|-------|-------|
| NCDs | 4 | 4 | | 8 |
| Infectious diseases | 11 | 10 | 2 | 23 |
| Health infrastructure and delivery | 44 | 14 | 1 | 59 |
| Research partnerships | 34 | 11 | 1 | 46 |
| **Total** | **93** | **39** | **4** | **136** |

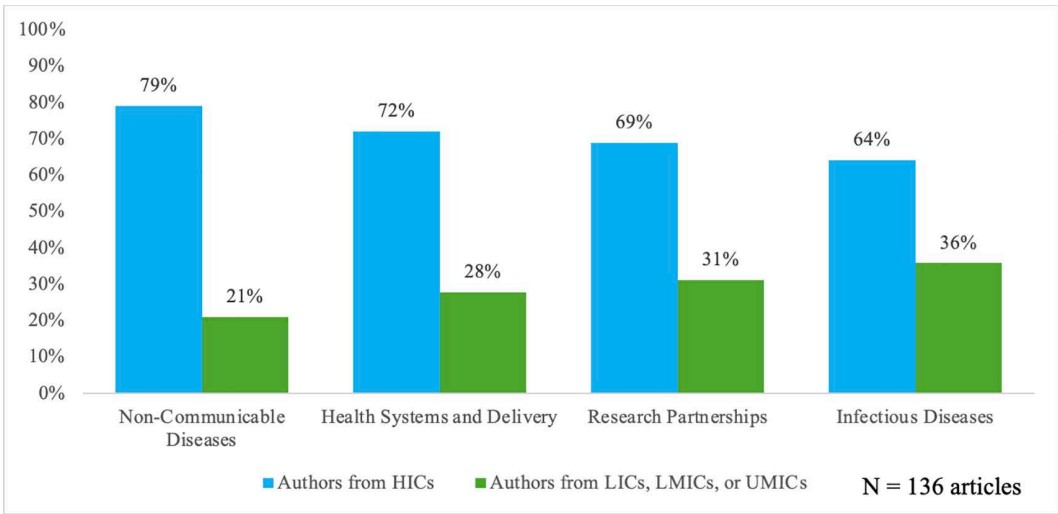

**Fig 2. Average regional representation of authors by classified article subject matter.**

institutions. Given that first and senior authors often lead the framing and writing and get the most visibility [30], we argue that representation from the Global South needs to be strengthened especially for first and senior authorship among commissioned content. We recognize that some authors could use their preferred affiliation in the Global North or the Global South depending on the topic, which affect our understanding of Global North and Global South representation [9]. Institutional affiliation does not always serve as a proxy for nationality, lived experience, or socio-cultural background [31]. As scholars have argued before, we also recognize that the framing based on Global South versus Global North and World Bank country income categories can mask the complexity of identity, lived experiences, and interconnected representation of a country and are imperfect [32,33]. However, even with these limitations in categorization, as indicated above there is scope for strengthening representation.

Additionally, it is important to note that across all subject matter categories, HIC institution affiliated authors representation is much higher than non-HIC authors as well (Fig 1). These results indicate a necessity to invite more commissioned content from authors in the Global South. Encouragingly, authors affiliated with institutions in the Global South were more strongly represented in articles that focused on LICs, LMICs, and UMICs, comprising on average 43% of authorship in these articles compared to an overall average of 30%. Therefore, PLOS GPH front matter content about the Global South was largely written by authors from the Global South.

Gender representation among commissioned content indicated higher representation of authors identifying as men versus women, across all authors (54% men vs 46% women). This is higher than the results indicated by a similar analysis of peer reviewed global health journals, which found 39.1% women authorship [29]. In our analysis, the higher representation of women among first authors (41% men vs 59% women) was an encouraging sign illustrating the results of intentional efforts of the journal editors to push for better gender representation. Senior authorship however, showed much higher representation of authors identifying as men (56%) compared to the representation of women (44%). This perhaps reflects the overall status quo in the global health landscape of senior authorship structurally favouring men. As a 2025 report by *Global Health 50/50* and *Lancet Commission on Gender and Global Health* indicated more than 70% of leaders in the global health sector are men [34]. Furthermore, another consideration for the editorial board is to commission articles from those who publicly identify as trans, non-binary, agender and genderfluid. In a climate where non cisgender identities are being attacked providing platforms for those whose voices are marginalized can contribute towards strengthening equity of knowledge production in global health [35,36]. Furthermore, based on available and analyzed authorship

data, there is space for the PLOS GPH to strengthen the commissioned content from those who publicly identify as Indigenous (currently 5% of first and 5% of senior authors publicly identify as Indigenous).

Considering the subject matter focus of articles, most of the articles have a global focus (78%) while all low- and middle-income country categories (UMIC, LMIC, LIC) have the second largest proportion of articles (36%). Considering the current political economy of global health and the necessity highlighted by many scholars and practitioners strengthen Global South leadership in global health [37–39], there is an opportunity for global health journals to platform views from the Global South on re-imagining global health. PLOS GPH has already shown leadership with intentional efforts in providing a space to Global South voices.

A key observation made during this study is the lack of data on authorship identities related to disability. Scholars have noted the persistence of the ableist culture even within the field of global health and field of medicine [40]. This could be a key reason for authors not publicly identifying as a person with disability. Recognizing this shortcoming, structural and intentional changes are needed in two aspects. First, journals including PLOS GPH, need to facilitate the option for authors to indicate their disability status as part of authorship data. Second, using the collected data journals should take steps to strengthen representation of persons with disabilities in their commissioned content. Furthermore, intersectionality of identity is difficult to capture using secondary data. Often intersectionality of marginalized identities can compound the limitations of access and opportunity to publish. Therefore, we believe further research which examines authorship identity in an in-depth manner, including intersectionality of identity and its impact, is necessary [41]. An inter-related additional focus for future research could also be how language limitations (English being the sole language in many journals) impact academic publication access and opportunity, in turn affecting author representation.

Overall, considering data availability, we recommend PLOS GPH to employ a proactive approach in their article submission process and provide the option for authors to indicate basic identification data such as race, geographic location, disability status, gender and other relevant data. Given safety and discrimination challenges, provision of this data by authors should be optional. However, having the option will encourage authors to declare their diverse identities, which in turn can help journals assess representation in authorship and if necessary, take steps to improve their authorship diversity and related topic diversity.

Finally, while applauding the self- reflection and subsequent equity focused changes made in journals such as *Lancet Global Health*, *Nature*, *Health Policy and Planning*, and other journals [13,18–20,42], we also recommend other journals to consider similar analyses of authorship representation, subject matter representation on content that they have editorial control over. We recommend these analyses made public and to share the subsequent strategic pathways forward.

## Limitations

First, we note that we used publicly available data and did not contact authors individually for this study. This approach may have limited gaining a complete understanding of author affiliation and representation. Second, authors may not share their full identity (e.g., identity as an Indigenous person or gender identity) in public-facing platforms due to fear of discrimination. Therefore, the results while providing an in-depth understanding of authorship representation and diversity, does have limitation in terms of providing exactly precise reflection of the diversity of authors contributing to PLOS GPH and their positionality. Third, we would also like to note that PLOS GPH only started commissioning essays from September 2024, contributing to the low number of essays available for analysis. This contributed to the lower sample size of essay articles. However, we also view this as an opportunity for the journal to strengthen representation further, using results of our analysis, in this new category of articles.

## Conclusion

PLOS GPH has made significant progress with their intentional efforts as a journal to strengthen equity in global health publication space. Although, the commissioned content of the journal still has a high representation of HIC-affiliated

authorship, as data indicate topics related to UMIC, LMIC and LIC are mostly written by authors from those countries. Gender representation of women and men is relatively even. However, there is space for including voices who publicly identify as transgender, non-binary, agender and gender-fluid. Given the current political economy in global health, PLOS GPH can have more commissioned content that explicitly focus on the Global South, especially as it pertains to making structural change in global health. Finally, while applauding the self- reflection and subsequent equity focused changes made in journals such as *Lancet Global Health*, *Nature*, *Health Policy and Planning*, and other journals [13,18–20,42], we also recommend other journals to consider similar analyses of authorship representation, subject matter representation on content that they have editorial control over. Continued self-assessment as journals can help measure progress and recognize strengths and shortcomings of equity focused efforts – leading to effective change in the long run.

## Supporting information

**S1 Table. Main focus areas of all articles analyzed listed under the four thematic categories.**
(DOCX)

## Acknowledgments

We express our sincere appreciation to Dr. Madhukar Pai and Ms. Julia Robinson for inviting us to do this assessment. We appreciate the transparency and support in providing access to the journal content and for answering our queries via meetings and emails. Finally, we thank and acknowledge Dr. Ananya Banerjee's input in relation to authorship representation of persons with disabilities in global health.

## Author contributions

**Conceptualization:** Esme Supriya Gupta Longley, Shashika Bandara.

**Data curation:** Esme Supriya Gupta Longley.

**Formal analysis:** Esme Supriya Gupta Longley, Shashika Bandara.

**Methodology:** Esme Supriya Gupta Longley, Shashika Bandara.

**Visualization:** Esme Supriya Gupta Longley.

**Writing – original draft:** Esme Supriya Gupta Longley, Shashika Bandara.

**Writing – review & editing:** Esme Supriya Gupta Longley, Shashika Bandara.

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
