## [Decision Letter · Decision Letter 0]

4 Jul 2025

PGPH-D-25-01321

Towards inclusive authorship: Analyzing author representation in PLOS Global Public Health front matter content

Dear Dr. Bandara,

Thank you for submitting your manuscript to PLOS Global Public Health. After careful consideration, we feel that it has merit but does not fully meet PLOS Global Public Health’s publication criteria as it currently stands. Therefore, we invite you to submit a revised version of the manuscript that addresses the points raised during the review process.

As an academic editor, I particularly appreciate this timely and relevant article that addresses important questions about equity and representation in academic publishing. Thank you for your work.

Here is a summary of the key feedback to address in your minor revision:

1. Temporal Analysis (Reviewer 2) - suggested change

The authors should leverage their 2021-2024 dataset to conduct a before/after comparison, specifically examining 2021-2022 versus 2023-2024 data. This would help assess whether PLOS Global Public Health's editorial commitment to diverse authorship has resulted in measurable increases in Global South representation. A basic statistical test should be applied to determine significance.

2. Critical Examination of Categorization Methods (Reviewer 3) - required change

The current approach of using World Bank income classifications and institutional affiliations as proxies for Global North/South status needs more nuanced discussion. The authors should acknowledge that this methodology may oversimplify complex individual positionalities, particularly for scholars who have Global South lived experiences but are affiliated with high-income country institutions. A more reflexive discussion of these limitations is needed to avoid essentializing identities based solely on institutional affiliation.

3. Intersectionality Framework (Reviewer 3) - required change

Rather than treating gender, income affiliation, and Indigeneity as separate categories, the authors should incorporate discussion of how these identities intersect and create compounded marginalizations. While data limitations may prevent full quantitative intersectional analysis, the authors should include a reflective section that conceptually addresses the significance of intersectionality and acknowledges the layered nature of structural exclusion in academic publishing.

4. Technical Revisions

All minor comments and suggestions from reviewers should be incorporated or responded to throughout the manuscript.

We look forward to receiving your revised manuscript.

Kind regards,

Marie Meudec, PhD

Academic Editor

Journal Requirements:

1. Please send a completed 'Competing Interests' statement, including any COIs declared by your co-authors. If you have no competing interests to declare, please state "The authors have declared that no competing interests exist". Otherwise please declare all competing interests beginning with the statement "I have read the journal's policy and the authors of this manuscript have the following competing interests:"

2. Please provide separate figure files in .tif or .eps format.

Reviewers' comments:

Reviewer's Responses to Questions

**Comments to the Author**

1. Does this manuscript meet PLOS Global Public Health’s publication criteria ? Is the manuscript technically sound, and do the data support the conclusions? The manuscript must describe methodologically and ethically rigorous research with conclusions that are appropriately drawn based on the data presented.

Reviewer #1: Yes

Reviewer #2: Yes

Reviewer #3: Yes

2. Has the statistical analysis been performed appropriately and rigorously?

Reviewer #1: Yes

Reviewer #2: Yes

Reviewer #3: Yes

3. Have the authors made all data underlying the findings in their manuscript fully available (please refer to the Data Availability Statement at the start of the manuscript PDF file)?

Reviewer #1: Yes

Reviewer #2: Yes

Reviewer #3: Yes

4. Is the manuscript presented in an intelligible fashion and written in standard English?

Reviewer #1: Yes

Reviewer #2: Yes

Reviewer #3: Yes

5. Review Comments to the Author

Reviewer #1: This article provides a less-researched but crucial asepct of research and gives interesting findings. The manuscript is well-explained and a scientific flow of information has been maintained. Overall, the article is interesting and has the potential to initiate dialogue.

Reviewer #2: Review

Thank you for the opportunity to review “Towards inclusive authorship: Analyzing author representation in PLOS Global Public Health front matter content”. This is really important work! Below are some suggestions to help clarify arguments.

MAJOR COMMENTS:

* Your data is from 2021-2024 and you noted that PLOS GPH had a 2-year review in which the editorial board committed to being more intentional about including diverse authorship. Perhaps consider comparing author affiliation income categories in 2021-2022 vs. 2023-2024 to see if proportion of Global South authorship increased, using a basic statistical test. This could help inform whether more can be done to promote Global South authorship.

MINOR COMMENTS:

Abstract:

* Change “It has contributed to epistemic injustice” to “This has contributed to epistemic injustice”

* Where you say “has had an impact on policymaking” - do you mean policymaking as a downstream result of studies published in PLOS GPH?

* In the first instance of “PLOS Global Public Health” put in brackets beside it “(GPH)”

* Remove the comma in “PLOS Global Public Health, was launched in 2021”

* Change “using genderize.io.” to “used genderize.io.”

* “and when unavailable, using genderize.io.” - in what percent of instances was it unavailable?

Introduction:

* Change “have been a long-standing challenge” to “have been long-standing challenges”

* Change “lack of journal policies on inclusive authorship, lack of support” to “lack of journal policies on inclusive authorship, and lack of support”

* Change “Epistemic injustice in global health knowledge production in turn influence global and national policies” to “Epistemic injustice in global health knowledge production, in turn, influences global and national policies”

* Change “Health Policy and Planning committing to at least 50% of articles include a first or senior author from a low and middle income country” to “Health Policy and Planning committed to including at least 50% of articles with a first or senior author from a low and middle income country”

* Change “In its two-year update, PLOS GPH editorial board” to “In its two-year update, PLOS GPH’s editorial board”

Methods:

* “Under authorship representation analysis we examined the country income category, gender, and Indigeneity of each author.” - How was group/consortium authorship dealt with?

* Change “we examined the country or region the article content is focused on” to “we examined the country or region the article content was focused on”

* “For first and senior authors, who often lead the writing and framing of commissioned opinion, review and essay articles we conducted an additional layer of analysis.” - after this sentence, you repeat the fact that you extract information on primary affiliations. I suggest removing that repeated information, and instead following this sentence with “In addition to primary institutional affiliations, for first and senior authors, we also examined the secondary ….”

* Change “Using the collected authorship data we did two types of calculations:” to “Using the collected authorship data we performed two different calculations:”

* By “b) the average of authorship based on country income category” do you mean “b) the average proportion of authorship based on country income category”?

* For gender and Indigeneity you note that you searched publicly available profiles - I suggest listing the exact online platforms that were searched, for reproducibility purposes. If there are many, perhaps include them in a supplementary table.

* Change “Supplementary File 1includes” to “Supplementary File 1 includes”

Results:

* Change “We considered all the authors (878), across all article types (136)” to “We considered all authors (n = 878) and articles (n = 136) in the following analyses.”

* Change “First (135) and senior (117) author represented across all commissioned articles” to “First (n = 135) and senior (n = 117) authors represented across all commissioned articles”

* Not sure if Table 3 is needed since all of the information is quite easy to summarise in text, as done in the paragraph above.

* “Using publicly available data and genderize.io we inferred gender identities across all 878 authors.” - for how many authors did you use genderize.io?

* Change “6 of the 117 senior authors (5%) self-identified as being Indigenous.” to “Six of the 117 senior authors (5%) self-identified as being Indigenous.”

* Be consistent with the use of “non-communicable diseases” vs NCDs - I suggest using the full-form in the first instance you mention it, then using NCDs afterwards.

Discussion:

* Change “PLOS GPH as of November 2023 had 50-50 Global South – Global North representation” to “As of November 2023, PLOS GPH had equal Global South and Global North representation”

* “HIC institution affiliated authors representation is much higher than non-HIC authors as well (Figure 4).” - do you mean Figure 1?

* Change “Therefore, in PLOS GPH front matter content about the Global South” to ”Therefore, PLOS GPH front matter content about the Global South”

* Remove “across all authors” from “Gender representation among commissioned content indicated higher representation of authors identifying as men versus women across all authors”

* Change “there is an opportunity for global health journal to platform views” to “there is an opportunity for global health journals to platform views”

* Under limitations - if the proportion of authors that required the use of genderize.io was high, this could also be a limitation

Reviewer #3: Comments to the Author

Dear Authors,

Thank you for the opportunity to review your important and timely manuscript, “Towards inclusive authorship: Analyzing author representation in PLOS Global Public Health front matter content.”

This is a carefully designed, transparent, and well-structured analysis exploring the equity, diversity, and inclusion practices within PLOS Global Public Health’s commissioned front matter content over a three-year period. Your study contributes meaningfully to ongoing discussions around epistemic justice, decolonization, and the democratization of knowledge production in global health scholarship.

I commend the authors for:

a systematic approach to data collection

clear use of publicly available data

a well-articulated discussion grounded in contemporary global health equity literature

recommendations that are specific and actionable for journals seeking to shift power in authorship

The writing is generally clear, well-organized, and accessible to a broad readership. The intention to assess your own journal’s progress is itself commendable and demonstrates a laudable commitment to transparency and accountability.

Below, I outline several suggestions that I believe will further strengthen the manuscript. These should be considered as constructive refinements rather than major overhauls.

Major Comments

Critical Reflexivity on Global North/Global South Categorization

While your use of World Bank income classifications is widely recognized, I strongly encourage a deeper reflection on the limitations of this framework. Institutional affiliation does not necessarily capture a researcher’s nationality, cultural background, or lived experience. For example, a Global South–born scholar with a HIC affiliation may still bring unique local knowledge, yet be coded here as Global North. A paragraph expanding on these limitations, including references to scholarship on “academic migration” and the politics of identity, would enrich the interpretation.

Intersectionality and Layered Inequities

Currently, the analysis treats gender, country affiliation, and Indigeneity separately. A deeper exploration of how these identities may intersect, for example, the experiences of Indigenous women authors or Global South women scholars, could greatly enhance the discussion. Even if intersectional quantitative data were not available, a conceptual reflection would be valuable, acknowledging the limitations of a univariate approach.

Treatment of Dual Affiliations

You note that authors were classified by their primary affiliation, with secondary affiliations recorded separately. Please clarify the rationale and potential biases that might emerge from this choice. Some authors list HIC primary affiliations while maintaining deep substantive partnerships with Global South institutions, which may not be reflected in the analysis. A clear justification and acknowledgment of this complexity would strengthen methodological transparency.

Limited Representation of Disability

You highlight the near-absence of disability data in your analysis. Consider moving this observation earlier in your discussion, to emphasize its significance. Given that disability justice is increasingly recognized as integral to equity, a clearer call for journals to collect optional disability identification data would be a powerful addition.

Language Barriers and Linguistic Hegemony

Language is a critical but often underappreciated determinant of epistemic injustice. English-language dominance in global health scholarship acts as a gatekeeping mechanism that disadvantages non-native speakers. While you mention structural barriers, I encourage you to explicitly reflect on language and its impacts on authorship diversity.

Minor Comments

Editorial Consistency

Check consistency in hyphenation: “Global-South-affiliated” appears with and without hyphens.

A few sentences are long and could be split for readability (for example, parts of the methods).

Terminology Clarifications

Please clarify “front matter” in the introduction for readers less familiar with journal publishing terminology.

When referencing “commissioned content,” you could provide a concise definition for non-editorial readers.

Data Availability Note

You mention an OSF repository. It would be helpful to explicitly restate the DOI or persistent link in both the methods and the data availability section to encourage reuse.

Visuals

Figures are helpful, but Figure 1 and Figure 2 could use slightly clearer axis labels. Consider larger font sizes for accessibility.

Check if color contrasts in the figures meet accessibility standards for color-blind readers.

Citations

References are comprehensive. A couple of recent works on decolonizing knowledge systems (for example, Tuck & Yang, 2012 Decolonization is not a metaphor) could be cited to situate your arguments even more strongly.

Overall Recommendation

Your manuscript is highly relevant, well-conceived, and rigorous. I recommend accept with minor revisions. The refinements suggested above would enhance the manuscript’s conceptual nuance and its contribution to the global conversation around equitable knowledge production.

I appreciate the opportunity to engage with your thoughtful and reflective scholarship and hope these comments prove useful as you finalize your paper.

Warm regards,

Meher Suri

6. PLOS authors have the option to publish the peer review history of their article (what does this mean? ). If published, this will include your full peer review and any attached files.

**Do you want your identity to be public for this peer review?** For information about this choice, including consent withdrawal, please see our Privacy Policy .

Reviewer #1: No

Reviewer #2: **Yes: ** Aashna Uppal

Reviewer #3: **Yes: ** Meher Suri

---

## [Editor Report · Decision Letter 1]

30 Jul 2025

Towards inclusive authorship: Analyzing author representation in PLOS Global Public Health front matter content

PGPH-D-25-01321R1

Dear Dr. Bandara,

We are pleased to inform you that your manuscript 'Towards inclusive authorship: Analyzing author representation in PLOS Global Public Health front matter content' has been provisionally accepted for publication in PLOS Global Public Health.

Thank you for revising the article based on the reviewers' comments. You have revised the article accordingly, incorporating the requested suggestions (classifications and intersectional framework) and technical revisions. As for the suggested modification (temporal analysis), your justification for not implementing it seems entirely valid.

Best regards,

Marie Meudec, PhD

Academic Editor
